# Engineering advanced logic and distributed computing in human CAR immune cells

Jang Hwan Cho[1,2,9], Atsushi Okuma[1,2,9], Katri Sofjan[1,2], Seunghee Lee[1,2], James J. Collins [3,4,5,6,7,8] &
Wilson W. Wong [1,2✉]

The immune system is a sophisticated network of different cell types performing complex biocomputation at single-cell and consortium levels. The ability to reprogram such an interconnected multicellular system holds enormous promise in treating various diseases, as exemplified by the use of chimeric antigen receptor (CAR) T cells as cancer therapy. However, most CAR designs lack computation features and cannot reprogram multiple immune cell types in a coordinated manner. Here, leveraging our split, universal, and programmable (SUPRA) CAR system, we develop an inhibitory feature, achieving a three-input logic, and demonstrate that this programmable system is functional in diverse adaptive and innate immune cells. We also create an inducible multi-cellular NIMPLY circuit, kill switch, and a synthetic intercellular communication channel. Our work highlights that a simple split CAR design can generate diverse and complex phenotypes and provide a foundation for engineering an immune cell consortium with user-defined functionalities.

[1] Department of Biomedical Engineering, Boston University, Boston, MA, USA. [2] Biological Design Center, Boston University, Boston, MA, USA. [3] Synthetic Biology Center, MIT, Cambridge, MA, USA. [4] Institute for Medical Engineering and Science, MIT, Cambridge, MA, USA. [5] Department of Biological Engineering, MIT, Cambridge, MA, USA. [6] Harvard-MIT Program in Health Sciences and Technology, Cambridge, MA, USA. [7] Broad Institute of MIT and Harvard, 415 Main Street, Cambridge, MA, USA. [8] Wyss Institute for Biologically Inspired Engineering, Harvard University, 3 Blackfan Circle, Boston, MA, USA. [9] These authors contributed equally: Jang Hwan Cho, Atsushi Okuma. ✉email: wilwong@bu.edu

A remarkable feature of the human immune system is its exceptional ability to sense and logically respond to diverse antigens and environmental signals. For instance, T cells have complex biocomputation circuitries that can detect antigens and integrate signals from co-stimulatory and co-inhibitory receptors in response to pathogens or tumors. Furthermore, the immune system can leverage specialized immune cell types to form consortia and perform distributed computing, where cells collectively address a challenge, with each cell type is tasked with sensing and producing a specific subset of inputs and outputs. In addition, immune cells can directly communicate with each other to attain temporally choreographed responses. The coordinated responses to infection from innate and adaptive immune cells through phagocytosis, cytotoxicity, and antibody generation highlight the sophistication of the distributed processing and communication features of the immune system. Such sophistication in computation and communication is required to achieve immune homeostasis and prevent diseases. Many therapies, especially antibody therapy, have been developed to modulate the sensing or cell–cell interactions of immune cells to treat a wide variety of diseases from autoimmunity to cancer[1–3]. However, many of these immunotherapies cannot discriminate targets based on the combination of multiple antigens nor involve various cell types, thus limiting their applicability.

The ability to engineer complex logic in human immune cells can greatly improve specificity and further unlock the potential of immunotherapy. Additionally, such a synthetic approach could provide insights into the governing principles of biocomputation in human immune cells. The chimeric antigen receptor (CAR), which is typically composed of a single chain variable fragment fused to signaling domains from T-cell receptor and co-stimulatory receptor, was developed to redirect T-cell specificity toward cancer cells, with very high efficacy against a few cancers[4–7]. Current conventional CAR designs, however, can only detect one antigen, and therefore have limited specificity. To improve specificity and control, several approaches have been explored to incorporate basic logic and control functions into CARs. For instance, CAR systems that are only active when two antigens are present on a cancer cell (AND gate) have been created[8,9], whereby T cells are transduced with a CD3ζ CAR directed towards one antigen and a chimeric co-stimulatory receptor directed towards a second antigen. In conventional T cells, activation of both CD3ζ and a co-stimulatory signaling pathway (e.g., CD28 or 4-1BB) are needed to induce the full response to eradicate tumors. However, the activity of these combinatorial CAR systems cannot be tuned once the cells have been engineered. Furthermore, for the combinatorial CAR design to function effectively as an AND gate, the activity of each CAR needs to be carefully balanced[8,9].

The ability to engineer distributed processing and cell–cell communication between human immune cells could lead to the development of synthetic immune cell consortia, which could further improve the safety and efficacy of cellular immunotherapy. Moreover, such a synthetic approach would provide insights into the governing principles of biocomputation in human immune cells. However, few studies have yet to combine multiple cell types in the immune cell engineering strategy.

Recently, we developed a split, universal, and programmable (SUPRA) CAR system to improve specificity and controllability[10]. The SUPRA CAR system is composed of a soluble antigen-binding portion, zipFv, and a universal signal transduction receptor, zipCAR, expressed on T cells. The zipFv has a leucine zipper and a single chain variable fragment (scFv). The zipCAR has intracellular signaling domains and an extracellular cognate zipper that specifically binds to the zipper on the zipFv. These zippers bridge the binding between the target antigen and

zipCAR-expressing T cells, and elicit T-cell responses. The SUPRA CAR system enables ON/OFF switching, fine-tuned T-cell activation, and AND logic computation. Moreover, the orthogonal SUPRA CARs can independently control different T-cell subsets. These functions highlight a powerful design feature afforded by the split CAR framework—namely, a collection of orthogonal split CARs controlling different signaling domains and/or expressed in different cell types can achieve complex biocomputation at the single-cell level and consortium level.

To expand the computational repertoire achievable with CARs in different immune cell types, we first introduce our SUPRA systems into seven distinct innate and adaptive immune cell types and implement tunable AND logic in a subset of them. Furthermore, we identify an inhibitory domain that is functional within our SUPRA CAR system, thus allowing a NOT logic. Using this NOT gate with two other zipCARs, we develop a CAR system that can perform three-input (A AND B) AND NOT C logic. Leveraging these advanced tools, we engineer a synthetic immune cell consortium and inducibly controlled the polarization of macrophages. We demonstrate multicellular distributed computing through regulatory T-cell-mediated suppression of conventional CD4+ T cells. We also create a direct cell–cell communication channel using a zipFv secretion system, and an intercellular AND logic circuit. Together, these wide-range applications illustrate the versatility of the SUPRA CAR system as a platform for engineering advanced logic and cell–cell interactions in human immune cells. The present work also provides a distinct perspective on how to achieve logic computation in immune cells that have never been explored before. The programmability of the SUPRA CAR system could serve as a foundation for creating synthetic immune cell consortia to treat diseases from cancer and beyond.

## Results

**SUPRA CARs can activate diverse adaptive and innate immune cell types.** Since conventional CARs are functional in many immune cell types, we also tested whether the SUPRA CAR system can redirect antigen specificity in various T-cell subtypes, NK cell, and macrophage. These cell types were chosen because of their potential therapeutic applications. Here we will focus on evaluating the induciblity and logic operation of the SUPRA CAR system by measuring population-level cell killing for cytotoxic cells and cytokine production for other immune cells. These responses were chosen because of their clinical importance. All immune cells were lentivirally transduced to express zipCARs and co-cultured with or without α-Her2 zipFvs in the presence of the Her2-expressing Nalm6 target cells. Consistent with our previous report[10], SUPRA CAR can efficiently induce target cell killing by CD8+ T cells (Fig. 1a). Furthermore, we differentiated naive CD4+ T cells to Th1 and Th2 cells to show that SUPRA CAR-expressing Th1 and Th2 cells can secrete IFN-γ and IL-4, respectively, when the corresponding zipFv was added (Figs. 1b, c, and Supplementary Fig. 1a). Additionally, regulatory T (Treg) cells are a unique subtype of CD4+ T cells that show various antigen-dependent immunosuppressive phenotypes, such as secretion of IL-10 and expression of CTLA-4[11], and have been strongly suggested to be clinically effective in treating autoimmune disorders[12]. A pan-T-cell activation marker, CD69, was upregulated in SUPRA CAR-containing primary human Treg cells when the corresponding zipFv was added (Fig. 1d, Supplementary Fig. 1b, and S1c).

In addition to adaptive immune cells, innate immune cells, such as γδ T cells, Natural Killer (NK) cells and macrophages, can also serve as the host for the SUPRA CAR system. γδ T cells are an innate version of T cells that express a unique T-cell receptor

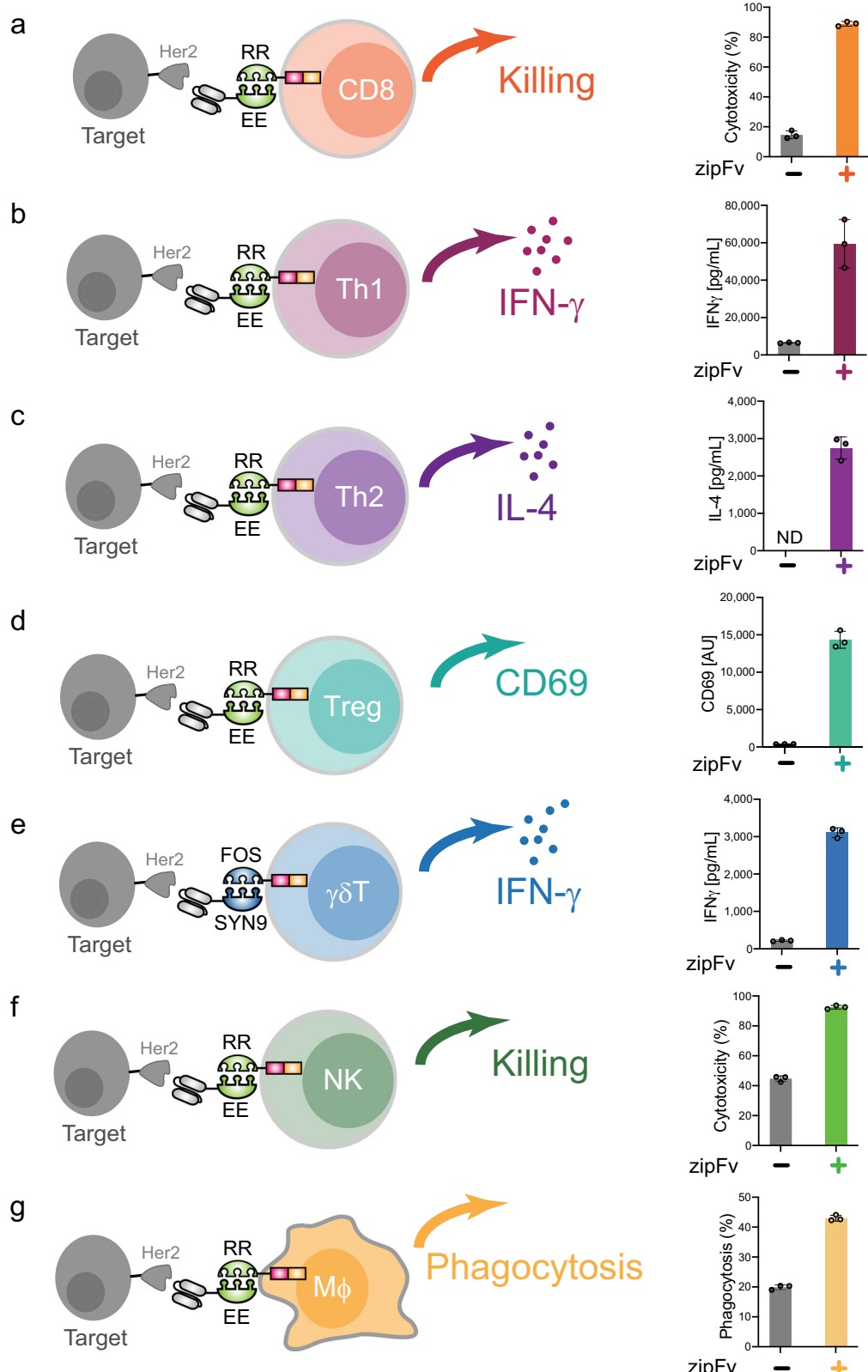

(TCR) composed of a γ-chain and a δ-chain. γδ T cells are a promising cell host for CAR immunotherapy against solid tumors because of their tissue-homing ability[13]. Moreover, γδ T cells can be obtained from healthy donors as an allogeneic cell source because the γδ TCRs are not specific to donor patient's proteins and therefore its allograft has quite low risk of graft-vs-host disease (GvHD)[14,15]. We showed that SUPRA CAR γδ T cells

secreted IFN-γ after activation by the administration of the corresponding zipFv (Fig.1e, Supplementary Fig.1d and 1e). NK cells are another type of lymphoid cells that are known to mediate anticancer effects without the risk of inducing GvHD[16]. NK-92, an established NK cell line, is also under investigation for adoptive immunotherapy applications and has demonstrated to be safe in phase I clinical trials[17–19]. We utilized an

**Fig. 1 The panel of cell types redirected by SUPRA CAR. a** Cytotoxicity by RR zipCAR-expressing CD8+ T cells. (Right) Nalm6 cells expressing Her2 were co-cultured in vitro with RR zipCAR-expressing CD8+ human primary T cells with and without α-Her2-EE zipFv ($n = 3$, data are represented as mean ± SD). **b** IFN-γ cytokine level from RR zipCAR expressing in vitro differentiated Th1 cells. (Right) Nalm6 cells expressing Her2 were co-cultured with RR zipCAR-expressing Th1 cells with and without α-Her2-EE zipFv ($n = 3$, data are represented as mean ± SD). **c** IL-4 cytokine level from RR zipCAR expressing in vitro differentiated Th2 cells. (Right) Nalm6 cells expressing Her2 were co-cultured with RR zipCAR-expressing Th2 cells with and without α-Her2-EE zipFv ($n = 3$, data are represented as mean ± SD). **d** CD69 expression level from RR zipCAR-FoxP3 expressing isolated Treg cells (CD4+ CD25hiCD127low). (Right) Nalm6 cells expressing Her2 were co-cultured with RR zipCAR-expressing Treg cells with and without α-Her2-EE zipFv ($n = 3$, data are represented as mean ± SD). **e** IFN-γ cytokine level from FOS zipCAR-expressing isolated γδ T cells. (Right) Nalm6 cells expressing Her2 were co-cultured with FOS zipCAR-expressing γδ T cells with and without α-Her2-SYN9 zipFv ($n = 3$, data are represented as mean ± SD). **f** Cytotoxicity by RR zipCAR-expressing NK-92MI cells. (Right) Nalm6 cells expressing Her2 were co-cultured in vitro with RR zipCAR-expressing NK cells with and without α-Her2-EE zipFv ($n = 3$, data are represented as mean ± SD). **g** Phagocytosis by RR zipCAR-expressing THP-1 macrophages. (Right) Nalm6 cells expressing Her2 were co-cultured in vitro with RR zipCAR-expressing THP-1 macrophages with and without α-Her2-EE zipFv ($n = 3$, data are represented as mean ± SD).

IL-2-producing NK-92 line, NK-92MI, and human primary NK cells[20], and established that the SUPRA CAR system can induce antigen-specific cytolysis (Figs. 1f, S1f). Macrophage is a type of myeloid immune cells and performs phagocytosis of opsonized pathogens and apoptotic cells. Through antigen presentation and cytokine release, macrophages control activation and differentiation of CD4+ helper T cells and immune responses. It has been reported that CD3ζ domain-containing conventional CAR induced macrophages to phagocyte antigen-expressing cells[21]. SUPRA CAR also redirected macrophage phagocytosis in a zipFv-dependent manner (Fig. 1g). Together, we showed the SUPRA CAR system can be used to controls diverse phenotypes from various immune cell types.

**SUPRA CARs can control macrophage polarization via CD4+ T-cell activation.** The activation of different immune cells, such as Th1 and Th2 cells, usually leads to different cytokines being produced, which can have a profound effect on other immune cells. Our orthogonal SUPRA CAR systems can regulate different subsets of immune cells independently, thus allowing us to control the type of immune cell to stimulate and the type of cytokines to produce. To demonstrate this function, we introduced orthogonal zipCARs into Th1 and Th2 cells. Th1 cells induce innate and adaptive immune cells to participate in cellular immunity via Th1 cytokines such as IFN-γ. Th2 cells induce innate and adaptive immune cells to participate in humoral immunity via Th2 cytokines such as IL-4. Macrophages polarize to proinflammatory M1 or anti-inflammatory M2 state in response to IFN-γ and IL-4 stimulation, respectively[22,23] (Supplementary Fig. 2a). Here, we co-cultured four cell types (1) antigens-expressing Nalm6, (2) RR zipCAR-expressing Th1 cells, (3) FOS zipCAR-expressing Th2 cells, and (4) THP-1-derived macrophages in vitro (Fig. 2a). The addition of α-Axl-EE zipFv to the cell mixture stimulated Th1 cells specifically to secrete IFN-γ and polarized macrophages to the M1 state, which expresses HLA-DR and CCR7 (Fig. 2b and Supplementary Fig. 2b). The addition of α-Her2-SYN9 zipFv stimulated Th2 cells to secrete IL-4 and polarized macrophage to the M2 state, which are CD206+ (Fig. 2b and Supplementary Fig. 2b). These data suggest that multiple types of zipCAR-expressing T cells can orthogonally and locally control the response of other immune cells.

**SUPRA CARs logically respond to combinatorial antigen in different cell types.** The lack of target specificity is one of the main challenges in CAR-T-cell therapy. Because many antigens that are overexpressed on cancer cells are also expressed on normal cells[24–26], identifying a unique cancer-specific antigen has proven to be challenging. To enhance tumor specificity,

combinatorial CAR systems against two antigens have been reported[8,27]. However, the logic function requires a fine balance of activity between two CARs with different signaling domains, which is very difficult to accomplish with the conventional fixed CAR design. To overcome this challenge, we previously developed a tunable AND logic circuit using orthogonal SUPRA CARs in human primary CD4+ T cells[10]. We now show that our AND logic circuit is also functional in primary CD8+ cytotoxic T cell (Fig. 3a and Supplementary Fig. 3a). We introduced the FOS zipCAR (binds to α-Her2-SYN9 zipFv) and the RR zipCAR (binds to α-Axl-EE zipFv) that contains either only the CD3ζ or co-stimulatory (CD28 or 4-1BB) domain, respectively[28,29]. Both types of engineered T cells showed synergistic upregulation of target cell killing with the addition of two zipFvs (Fig. 3b, c) as well as CD69 expression on T cells (Supplementary Fig. 3b). Compared to CD28, 4-1BB expressing AND gate CAR system showed higher basal activity when the 4-1BB signaling domain was activated. However, CD28 zipCAR-bearing cells performed AND logic in a wider range of zipFv concentrations than that of 4-1BB containing AND logic circuit.

Another cell type that can benefit from more precise targeting is regulatory T cell (Treg). Treg is a promising therapeutic agent to treat diverse inflammatory diseases including autoimmune disorders[12,30]. However, systemic immune suppression by polyclonal Treg will limit its therapeutic potential[31]. To enhance the specificity of Treg cells, we installed the AND gate system into Tregs by expressing the SYN6 zipCAR (binds to α-Axl-SYN5) containing CD3ζ domain and SYN1 zipCAR (binds to α-Her2-SYN2) containing CD28 co-stimulatory domain. We measured the CTLA-4 expression, which is an activation marker that plays an important role in immunosuppression[32], and demonstrated that this CD3ζ/CD28 AND logic is functional in Treg cells (Fig. 3d and Supplementary Fig. 3c). Together, the combination of CD3ζ and CD28 signaling domains can act as a tunable intracellular AND gate in multiple clinically relevant cell types.

**BTLA-derived co-inhibitory signaling domain can perform NOT logic.** We sought to develop a NOT gate in CAR-T cells (Fig. 4a) to increase the type of computation achievable beyond AND logic with CARs. Previously, PD-1 and CTLA-4 have been utilized in inhibitory CARs[27], but there are other co-inhibitory receptors[33] that may be leveraged for the NOT gate design. We generated a collection of zipCARs based on well-known co-inhibitory receptors (PD-1, LAG3, TIM3, BTLA, CTLA-4). CD4+ T cells and CD8+ T cells were transduced with a FOS zipCAR that contains CD28 and CD3ζ signaling domains and a RR zipCAR with different inhibitory domains (Supplementary Fig. 4a). Activation of the zipCAR with B and T lymphocyte associated (BTLA) strongly inhibited IFN-γ secretion in CD4+ T cells (Fig. 4b and Supplementary Fig. 4b). To confirm

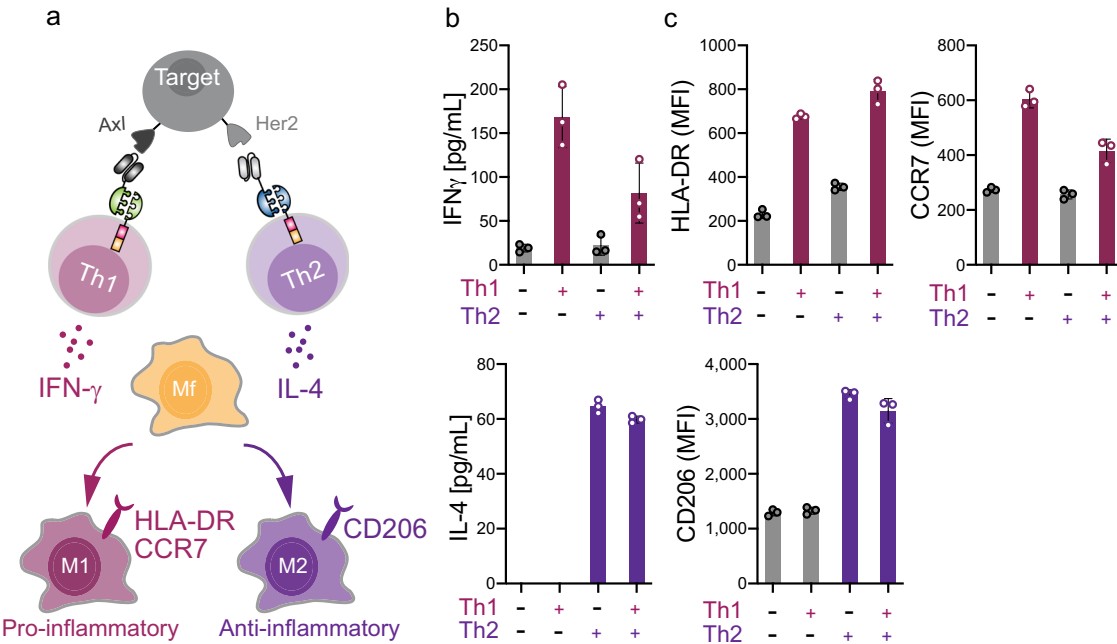

**Fig. 2 Engineering endogenous immune system with SUPRA CAR-expressing different T-cell subtypes. a** Schematic of controlling macrophage polarization by zipCAR-expressing Th1 and Th2 cells. The RR zipCAR and FOS zipCAR control activity of Th1 and Th2 cells, respectively. α-Axl-EE zipFv binds to RR zipCAR and activates Th1 cells. α-Her2-SYN9 zipFv binds to the FOS zipCAR and activates Th2 cells. Activation of Th1 and Th2 CD4+ T cells leads to secretion of IFN-γ and IL-4, respectively. Macrophage polarizes to M1 (proinflammatory) when exposed to IFN-γ secreted by Th1 cell and it polarized to M2 (anti-inflammatory) when exposed to IL-4 secreted by Th2 cells. **b** IFN-γ (Top) and IL-4 (Bottom) production from RR zipCAR-expressing Th1 cells and FOS zipCAR-expressing Th2 cells with or without 5 nM α-Her2-SYN9 zipFv and 5 nM α-Axl-EE zipFv (n = 3, data are represented as the mean ± SD). **c** HLA-DR (Top left) and CCR7 (Top right) expression levels in THP-1 macrophages were measured by flow cytometer 24 h after staring co-culture. (Bottom) CD206 in THP-1 macrophages was also analyzed at the same time as detecting M2 marker (n = 3, data are represented as the mean ± SD).

whether the inhibition of the BTLA domain is not induced by the simple addition of the corresponding zipFv, we co-cultured CD4+ T cells transduced with FOS-CD28-CD3ζ and RR-BTLA with various types of antigen-expressing target cells (Her2+ and Her2+Axl+, Supplementary Fig. 4e). As expected, the zipCAR with BTLA suppressed the IFN-γ secretion only in the presence of Axl+ target antigen. Furthermore, BTLA stimulation weakly suppressed IFN-γ secretion in CD8+ T cells (Supplementary Fig. 4c). However, in concurrence with the previous report[34], BTLA stimulation by zipFv addition did not suppress cytolysis activity in CD8+ T cells (Supplementary Fig. 4d).

We next examined whether the NOT gate can suppress target cell killing by NK cells. To identify ideal activation domains that can activate NK cell cytolysis and be suppressed by BTLA, we tested different NK cell-activating domains (CD3ζ, CD28-CD3ζ, NKG2D, 2B4, DAP12, CD28, and ICOS) and selected four domains (CD3ζ, CD28-CD3ζ, 2B4, and DAP12) that induced potent cell killing (Fig. 4c and Supplementary Fig. 5a). Then, we tested whether BTLA stimulation can inhibit cell killing induced by each activation domain (Supplementary Fig. 5b). BTLA significantly reduced the killing efficiency of CARs containing CD3ζ or 2B4 alone (Fig. 4d). Next, we varied the concentration of α-Her2-SYN9 and α-Axl-EE zipFvs to determine a range of zipFv concentrations that endows NOT logic function in NK cells (Fig. 4e). At the optimal dose of zipFvs, 2B4 containing NOT gate suppressed the killing of target cells by more than 40% when BTLA was activated. These results illustrate the design criteria for engineering a NOT gate; the selection of co-inhibitory domains needs to be balanced with the proper activating domains to achieve optimal NOT gate function and the choices of signaling domains will be cell-type dependent.

We also tested the NOT gate function in vivo using a human xenograft tumor model. We injected luciferase- and Her2-expressing SKOV3 ovarian cancer cell line orthotopically into the intraperitoneal cavity of *Prkdc^scid^Il2rg^tm1Wjl^*/SzJ (NSG) mice (Fig. 4f). After tumor engraftment, we infused CD3ζ/BTLA NOT gate expressing NK cells at day 1 and day 8. α-Her2-EE zipFv and α-Her2-SYN9 zipFv were administrated for 3 weeks to activate CD3ζ and BTLA, respectively. To eliminate the effect of competitive binding to Her2 between α-Her2-EE zipFv and α-Her2-SYN9 zipFv, we also injected α-Her2 scFv without any leucine zipper in the group of CD3ζ activation only. Tumor burden was measured by in vivo bioluminescent imaging at day 22 after tumor injection (Fig. 4g, Supplementary Fig. 5c and 5d). Activation of the chimeric receptor having CD3ζ signaling domain accelerated tumor killing by NK cells, which have substantial antitumor effects even without the chimeric receptor signaling as previously reported[35]. However, activation of BTLA significantly reduced the cytotoxicity and increased tumor burden in the mouse xenograft model. These results suggest that NOT logic with BTLA can be remarkably useful to improve the safety of the CAR therapy by identifying the combination of 2 antigens.

**3-input logic gate in single cell using the SUPRA CAR system.** To enhance logical programmability of engineered immune cells, we next tested if we can combine the AND and NOT logic into a single cell. Based on previous screening in Jurkat T cells, we found SYN1-SYN2, SYN6-SYN5, and FOS-SYN9 pairs to be compatible as three orthogonal inputs[10]. As expected, SYN1 zipCAR, SYN6 zipCAR, and FOS zipCAR-expressing CD8+ T cells killed all target cells in the presence of SYN2 zipFv, SYN5 zipFv, and SYN9 zipFv, respectively. Importantly, minimal cross-reactivity with the

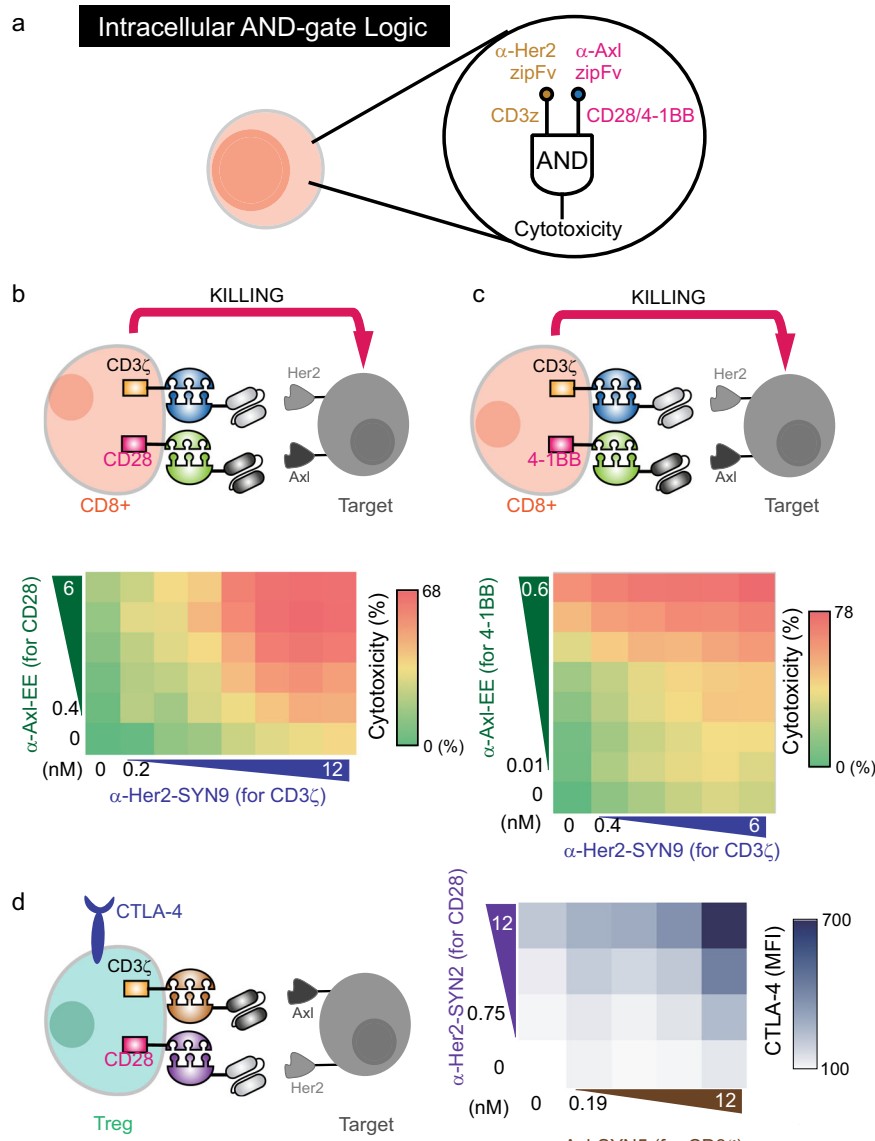

**Fig. 3 The intracellular AND logic with different signaling domains. a** Diagram of intracellular AND logic. **b** Primary human CD8+ T cells were transduced with FOS zipCAR-containing CD3ζ domain and RR zipCAR-containing CD28 domain. Cytotoxicity against Her2- and Axl-expressing Nalm6 was measured 24 h after adding α-Her2-SYN9 and/or α-Axl-EE zipFvs. The heatmap indicates cytotoxicity at varying zipFv concentrations ($n = 3$, data are represented as mean). **c** Cytotoxicity of CD8+ T cells transduced with FOS zipCAR-containing CD3ζ domain and RR zipCAR-containing 4-1BB domain. The heatmap indicates cytotoxicity at varying zipFv concentrations ($n = 3$, data are represented as mean). **d** (Left) Isolated Treg cells were transduced with two zipCAR constructs: SYN6-CD3ζ-P2A-FOXP3 and SYN1-CD28-P2A-puro. After puromycin selection (2 μg/mL), Treg cells were co-cultured with Her2- and Axl-expressing Nalm6 target cells (Right) The heatmap shows surface CTLA-4 expression detected after 48 h by flow cytometry at varying zipFv concentrations (α-Axl-SYN5 and α-Her2-SYN2) ($n = 3$, data are represented as mean).

other zipFvs was observed (Supplementary Fig. 6a). We next generated T cells expressing three zipCARs simultaneously, each contains either a CD3ζ, CD28, or BTLA domain (Supplementary Fig. 6b). We used dual CAR expression system that one vector containing 2 A ribosomal skipping site achieved two SUPRA CARs expression simultaneously (Supplementary Fig. 6c). The other vector has third SUPRA CAR and puromycin resistant gene to eliminate untransduced cells. Compared to activation of CD3ζ or CD28 signaling domain alone, triggering both CD3ζ and CD28 domains led to a significant increase in IFN-γ production (AND logic) (Fig. 5a). Furthermore, this IFN-γ upregulation was significantly reduced with BTLA stimulation. To demonstrate its generalizability in different cell types and to identify the range of zipFv concentration that confers logic capability, we also

transduced CD4+ T cells with triple zipCARs (Fig. 5b). In the 3-D dose response plot, the X–Y plane shows dual antigen sensing functionality from CD3ζ and CD28. As expected, AND logic represented synergistic IFN-γ upregulation (X–Y plane). In addition, BTLA activation strongly inhibited IFN-γ production in a zipFv dose-dependent manner (Z-axis). Our results demonstrate that 3-input multilogic in a single T cell is achievable in multiple cell types using the SUPRA CAR platform.

**SUPRA CARs can redirect Treg and Tconv responses for simultaneous and logical control of immune activation and suppression.** A hallmark of many biological processes, including those involved in the immune system, is coordination between different cells to perform distributed computation.

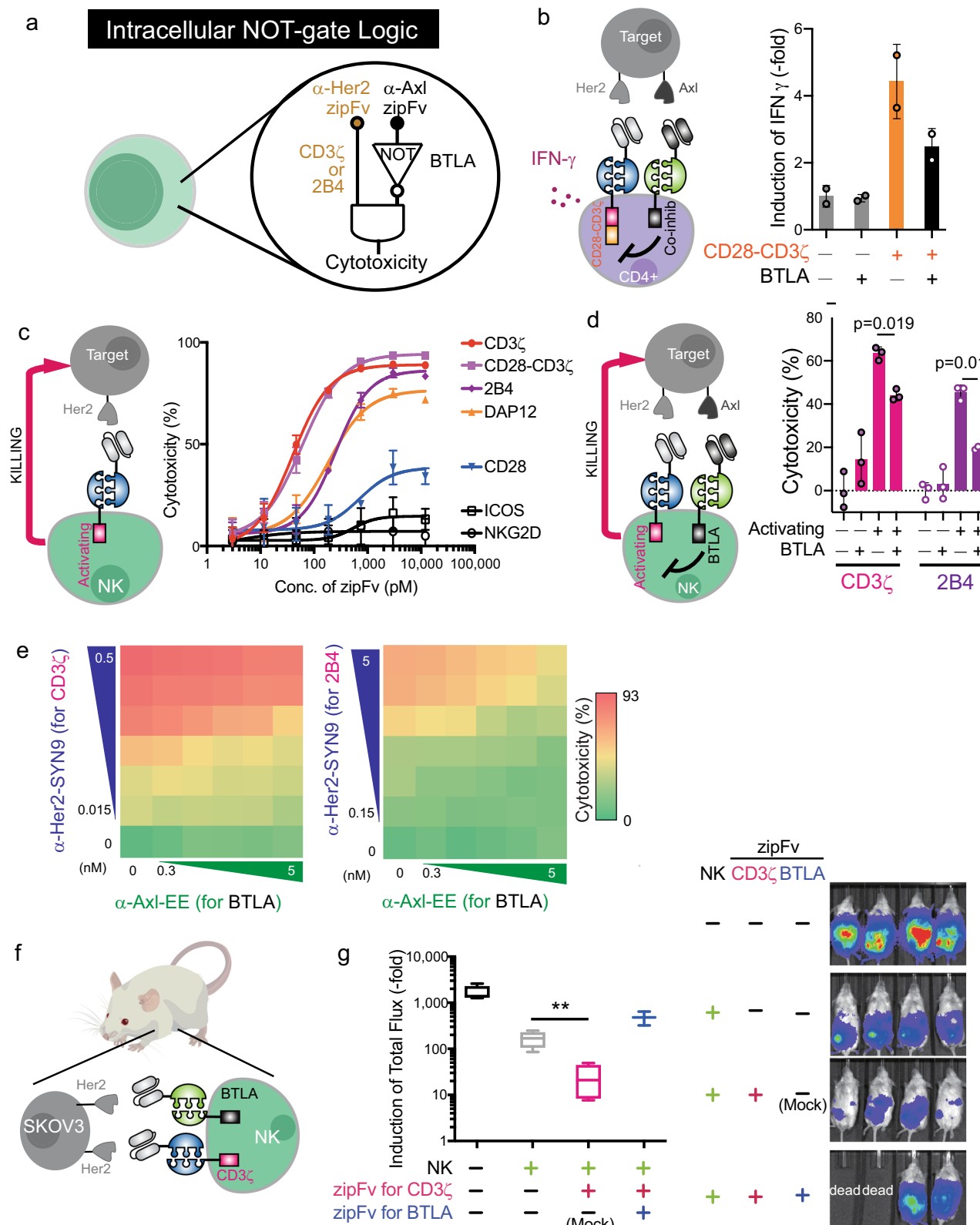

Given that our SUPRA CAR system is active in both conventional and regulatory T cells, we now have the ability to logically regulate inflammation and immune suppression in an inducible and antigen-dependent manner. To demonstrate this capability, we co-cultured FOS zipCAR-expressing conventional CD4+ T cells with RR zipCAR-expressing Treg cells in the presence of target cells that express Her2 and Axl (Fig. 6a).

When we added α-Her2-SYN9 zipFv to activate the FOS zipCAR expressed on Tconv cells, 40% of Tconv cells proliferated (Fig. 6b and Supplementary Fig. 7a). However, activating RR zipCAR-expressing Treg cell by α-Axl-EE zipFv significantly inhibited the proliferation of Tconv cells.

Furthermore, we tested the growth suppression by SUPRA CAR Treg cells utilizing the 2-input AND gate already shown in

**Fig. 4 The intracellular NOT logic with BTLA in different cell types. a** Diagram of intracellular NOT logic with BTLA co-inhibitory signaling domain. **b** IFN-γ production from CD4+ T cells transduced with FOS-CD28-CD3ζ and RR zipCAR with BTLA co-inhibitory domain. (Right) Supernatants were collected 24 h after adding 1.2 nM α-Her2-SYN9 zipFv and/or 12 nM α-Axl-EE zipFv ($n = 2$; data are represented as the mean + SD). **c** Effect of concentration of α-Her2-SYN9 zipFv on cytotoxicity performed by FOS zipCAR-expressing NK-92MI cells with various activation domains (magenta, CD3ζ; light purple, CD28-CD3ζ; dark purple, 2B4; yellow, DAP12; blue, CD28; black with square, ICOS; black with circle, NKG2D; $n = 3$; data are represented as the mean ± SD). **d** Suppression of cytotoxicities by BTLA. NK-92MI cells expressing FOS zipCAR with activating domains (CD3ζ or 2B4) and RR zipCAR with BTLA co-inhibitory domain were co-cultured with Her2 and Axl-expressing Nalm6 target cells in the presence of different combinations of zipFvs (α-Axl-SYN9, α-Her2-EE). (Right) Live target cells were measured by flow cytometry 24 h after co-culture (magenta, CD3ζ; dark purple, 2B4; $n = 3$; data are represented as the mean + SD; the statistical significance was determined by two-tailed student's t-test). **e** Effect of concentration of α-Her2-SYN9 and/or α-Axl-EE zipFvs on cytotoxicity performed by NK-92MI cells. SUPRA CAR-NK cells express FOS zipCAR with BTLA co-inhibitory domain and RR zipCAR with CD3ζ domain (Left) and with 2B4 activation domain (Right) ($n = 3$, data are represented as the mean). **f** Schematic of the xenograft mouse model for verification of NOT gate NK cells. NK-92MI cells were expressing both FOS zipCAR with CD3ζ and RR zipCAR with BTLA. Target SKOV3-luc cells have high-level Her2 expression. We also administrated α-Her2-EE zipFv, α-Her2-SYN9 zipFv, and α-Her2 scFv (Mock) to specified groups in Fig. 4g. **g** (Left) Box and whiskers graph shows tumor burden in each group at 22 days after tumor injection; tumor only without NK cells (black), tumor and NK cells without zipFv (gray), tumor and NK cells with α-Her2-EE zipFv and α-Her2 scFv (magenta), and tumor and NK cells with α-Her2-EE zipFv and α-Her2-SYN9 zipFv (blue). Each zipFvs were injected intraperitoneally every day for 21 days. (Right) Representative bioluminescence images at day 22. Box plots indicate median (middle line), 25th, 75th percentile (box), and 5th and 95th percentile (whiskers). $n = 4$ (4th group is $n = 2$ because 2 mice died before imaging); mean ± SD; the statistical significance was determined by Multiple t-test; **$p = 0.01$.

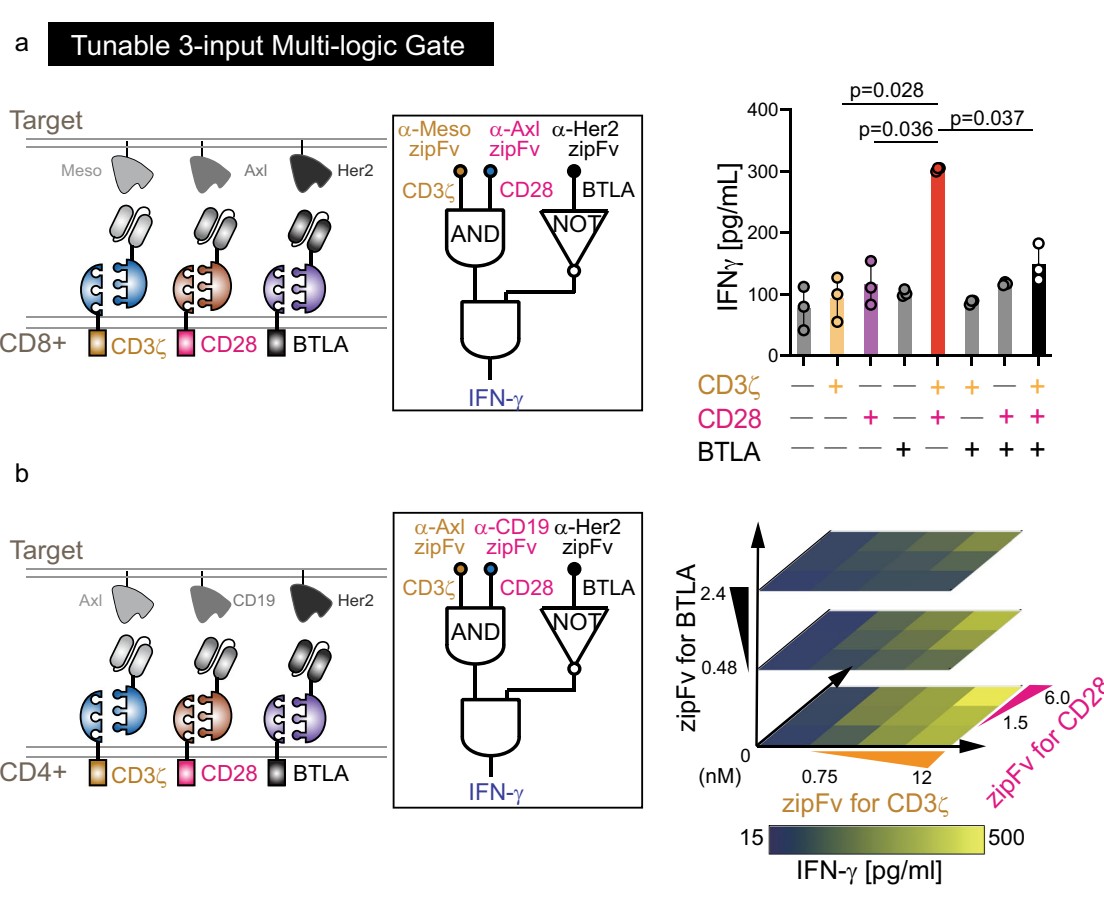

**Fig. 5 Tunable 3-input multilogic in the single cell. a** Design of orthogonal SUPRA CARs that control CD3ζ, CD28, and BTLA signaling domains inducibly and independently. Primary CD8+ T cells were engineered to express FOS zipCAR, SYN6 zipCAR and SYN1 zipCAR that contain CD3ζ domain, CD28, and BTLA signaling domain, respectively. In addition, α-Meso-SYN9 zipFv, α-Axl-SYN5 zipFv, and α-Her2-SYN2 zipFv lead to activation of CD3ζ, CD28, and BTLA, respectively. (Right) IFN-γ secretion was measured after co-culturing with Her2, Axl, and Meso expressing Nalm6 target cells with different zipFv combinations ($n = 3$, data are represented as the mean + SD, the statistical significance was determined by Student's t-test). **b** Primary CD4+ T cells expressing FOS-CD3ζ, SYN6-CD28, and SYN1-BTLA were co-cultured with Her2, Axl, and CD19 expressing Nalm6 target cells. The 3D heatmap shows IFN-γ production from 3-input CD4+ T cells at varying concentrations of three different corresponding zipFvs ($n = 2$, data are represented as the mean).

Fig. 3d. We co-cultured CD19 + Her2+Axl+ target cells, FOS zipCAR-expressing conventional CD4 T cells, and Treg cells expressing both SYN1-CD28 zipCAR and SYN6-CD28 zipCAR (Fig. 6c). When we added α-Axl-SYN9 zipFv, FOS zipCAR-expressing conventional CD4 + T cells proliferated (Fig. 6d and Supplementary Fig. 7b). Stimulation of either only CD28 or CD3ζ signaling domains from Tregs only weakly suppressed the growth of conventional CD4+ T cells. However, activation of both domains from Tregs strongly diminished the growth of conventional CD4+ T cells. This system acts as an inducible

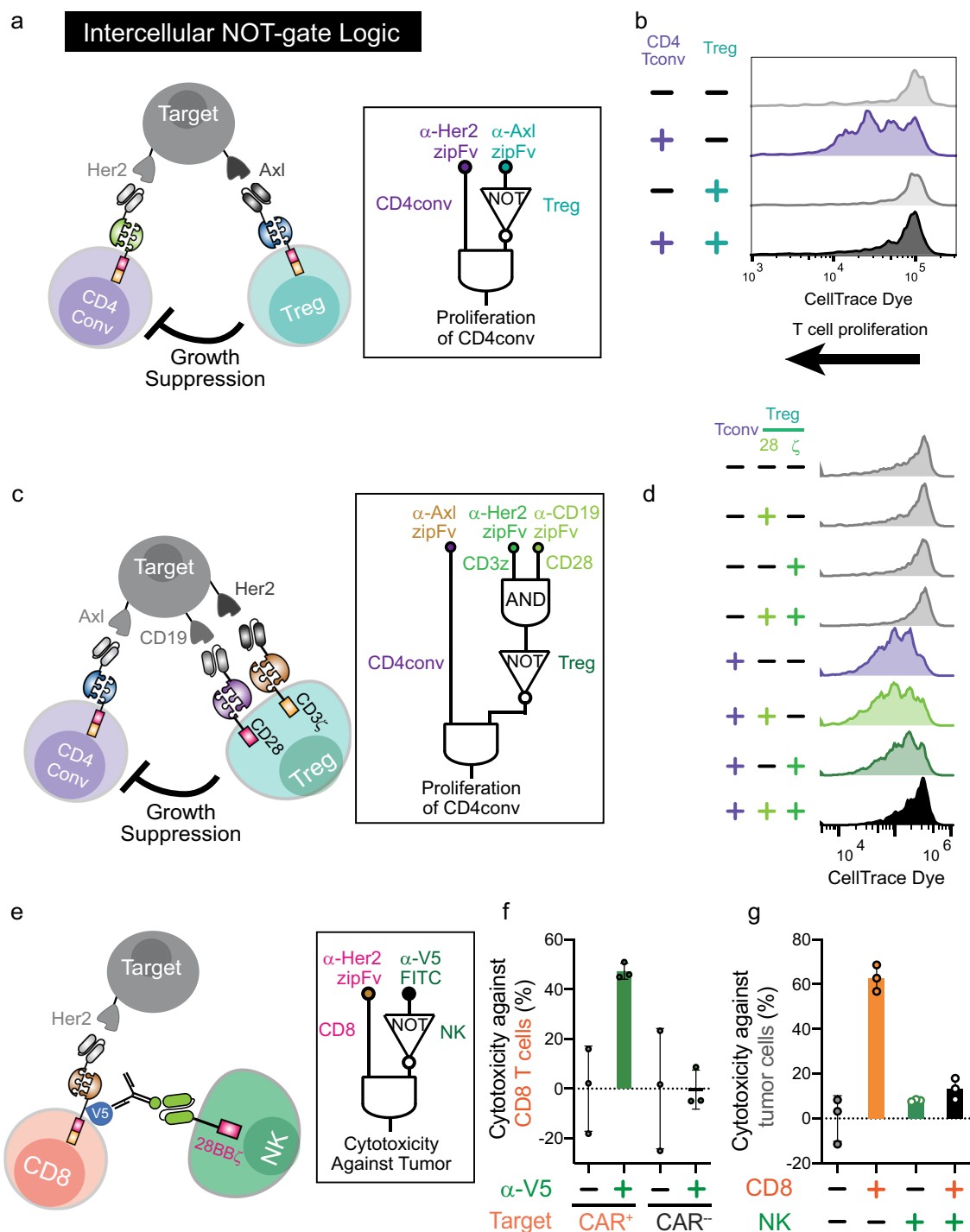

multicellular A AND NOT (B AND C) logic, which can be achieved only through the SUPRA CAR platform.

**A multicellular kill switch system.** The ability for different immune cells to regulate the survival of each other is an important feature of the immune system, as it affords homeostasis and prevents the deleterious effect of an overactive immune response. We sought to endow an engineered immune system with such features by directing cell killing toward our engineered immune cells in a specific and programmable manner. In particular, we generated a CAR-positive cell-specific killing system by targeting a V5 epitope tag on the zipCAR. Then, we added a FITC-conjugated α-V5 antibody and NK cells expressing an α-FITC-

CAR[36,37]. The α-V5-FITC antibody binds to the zipCAR CD8+ T cells, which subsequently recruit and activate the α-FITC-CAR-NK cells to eliminate zipCAR-expressing CD8+ cells (Fig. 6e). As expected, the number of zipCAR-expressing CD8+ T cells decreased in the presence of α-V5-FITC antibodies, while zipCAR-negative CD8+ T cells were not affected (Fig. 6f and Supplementary Fig. 7c). Furthermore, the number of tumor cells increased significantly, as NK cells prevent zipCAR CD8+ T cells from killing the tumor cells even in the presence of corresponding zipFvs (Fig. 6g). These data strongly suggest that this system can eliminate engineered cells and act as an alternative orthogonal kill switch, which can be useful to manage CAR-T-cell toxicities including cytokine release syndrome and off-tumor cytotoxicity.

**Fig. 6 The intercellular NOT gate with regulatory T (Treg) cells. a** Diagram of intercellular NOT gate with Treg cells. The RR zipCAR and FOS zipCAR control the activity of Treg and conventional CD4+ T cells (Tconv), respectively. α-Axl-EE zipFv binds to the RR zipCAR and activates Treg cells. α-Her2-SYN9 zipFv binds to the FOS zipCAR and activates CD4+ Tconv cells. Activation of Treg cells led to the suppression of CD4+ Tconv cells. **b** Suppression of growth of CD4+ Tconv cell by SUPRA CAR equipped Treg cells. CD4+ Tconv cells expressing FOS zipCAR were prelabeled with CellTrace Violet dye. When activated by CAR signaling, the fluorescence intensity of labeled cells decrease with cellular growth. Cells were analyzed by flow cytometer after 4 days of the co-culture period. The left shift of peaks indicates T-cell proliferation. Each plot shows dye fluorescence of CD4+ Tconv cells with different zipFv combinations (representative of three biological replicates). **c** Diagram of A AND NOT (B AND C) logic gate with AND gate Treg cells. Target Nalm6 cells express Her2, Axl, and CD19. CD4+ Tconv cells expressing FOS zipCAR were prelabeled with CellTrace Violet dye. Treg cells expressed SYN1-CD28 and SYN6-CD3ζ (see also Supplementary Fig. 3b). **d** Suppression of CD4+ conventional T-cell growth by AND gate Treg cells. α-Her2-SYN9 zipFv activated CD4+ conv T cells. α-CD19 SYN2 zipFv and α-Her2-SYN5 zipFv activated CD28 and CD3ζ in Treg cells, respectively. Histograms indicate the divided cells in CD4+ conv cells measured by fluorescent dye dilution for 4 days. **e** Diagram of the kill switch with CAR-targeting NK cells. Target NALM6 cells expressed Her2. CD8+ T cells and NK cells expressed SYN6-V5 tag zipCAR and α-FITC-CAR, respectively. **f** Cytotoxicity against CD8+ cells. α-V5-FITC was added to co-culture of CD8+ T cells and α-FITC-CAR-NK cells. The graph compares the cytotoxicity against SYN6-V5 zipCAR+ CD8+ T cells to CAR negative CD8+ T cells ($n = 3$; Mean + SD). **g** Cytotoxicity against target cells. α-Her2-SYN5 and/or α-V5-FITC were added to co-culture of SYN6-V5 zipCAR+ CD8+ T cells, α-FITC-CAR-NK cells, and target Nalm6 cells simultaneously. Live target cells are counted by flow cytometer 24 h after co-culture ($n = 3$; Mean + SD).

**Distributed computing with a cell–cell communication channels in immune cell consortia.** Cell–cell communication between immune cells is vital to coordinate a proper immune response. We want to explore how intercellular communication can be harnessed to achieve logical computation. Unlike other split CAR designs, all the components of the SUPRA CAR system can be genetically encoded. Therefore, we can control the expression of zipFvs in response to immune cell activation, which enables us to create an orthogonal communication channel between the engineered immune cells. Here, we utilized an NFAT promoter, which is activated in a TCR activation-dependent manner[38,39], to secrete a zipFv for activation of another zipCAR-expressing cells (Fig. 7a). In this setting, sender CD4+ T cells contain a (1) SYN6 zipCAR and (2) secretion module (NFAT promoter driving α-Axl-SYN2 zipFv). Receiver CD4+ T cells express a SYN1 zipCAR. Thus, after α-Her2-SYN5 zipFv was added to activate sender cells, α-Axl-SYN2 zipFv was secreted by sender cells and activated receiver cells, as measured by CD69 expression on both receiver and sender cells (Fig. 7b). However, when sender CD4+ T cells are devoid of the NFAT promoter-zipFv module, activation of the sender cell did not result in activation of receiver cells. These data showed that SUPRA CARs can perform an intercellular AND logic through a stimulus-inducible zipFv secretion system.

## Discussion

The immune systems are capable of sensing diverse antigens, performing complex computation based on input signals, and producing a wide range of responses. Understanding how to program these complex functions would have an enormous impact on medicine. Here, we demonstrated that SUPRA CAR system can be used with seven different immune cell types and orchestrate an inducible logic response from multiple cell types. The development of an inhibitory zipCAR allows us to generate the first 3-input logic circuits based on CARs in primary human immune cells. Furthermore, because our zipFv is genetically encodable, we can engineer T cells to secrete zipFvs, thus creating a synthetic cell–cell communication channel that can activate nearby corresponding zipCAR-expressing cells and generate a distributed logic circuit. Together, we showed the scalability of SUPRA CAR in synthetic immune cell consortia beyond the canonical CAR-T-cell therapy.

A major challenge facing current CAR-T-cell design, especially for solid tumors, is to identify a target that is uniquely expressed on the tumor but not on any vital organs. Combinatorial antigen recognition can help alleviate this challenge by increasing the number of suitable antigens, which are not specific by themselves,

to increase tumor specificity. In fact, several antigen pairings had been predicted as optimal CAR targets for acute myeloid leukemia[40]. Compared with previous logic gates[8,27], our AND gate in CD8+ T cells and NOT gate in NK cells can result in lower basal killing simply by adjusting each signal intensity with optimal concentrations of zipFvs. However, we recognize that our work, through our measurement of bulk immune response, is focused on the population-level logical performance, and the full immune response (e.g., various cytokines production level) at the single-cell level remains to be explored. Our data illustrate an important design principle that signal strength tunability is a key parameter for optimal logic performance. A split design system like SUPRA represents one of the most accessible approaches. Complex 3-input logic with CARs demonstrated here in human primary T cells would be very difficult, if not impossible, to achieve using other approaches to tune CAR activity, such as through controlling the expression levels of the CARs or varying affinities between scFv and tumor antigens.

We showed that NK cell is a better host than CD8+ T cell for actuating the NOT gate with the BTLA inhibitory domain (Fig. 4d). We speculate that NK cells may be more prone to inhibitory signaling because, unlike CD8+ T cells, NK cells constantly recognize ligands of inhibitory receptors to rapidly distinguish between self and non-self, and inhibitory signaling is dominant in NK cells[41]. In addition, the activating signaling from CD28-CD3ζ CAR, which is usually used for inducing full activation of T cells, was not inhibited by BTLA even in NK cells (Supplementary Fig. 5b). The number of ITAM domains in CD28 and CD3ζ showed functional differences in CAR-T cells[4], suggesting that the activating signaling from both CD3ζ and CD28 could be too strong for the NOT gate to be efficient. Therefore, in the future, we may need to find activating domains that can induce strong cytotoxicity but can be suppressed by inhibitory domains for better performance. Because CAR-NK cell does not require any co-stimulatory signal for full activation[42], we were able to utilize CD3ζ or 2B4 alone as an activating domain in NOT gate in NK cells. Cell types and signaling domain compatibility is another design criteria that need to be addressed when engineering CAR based circuits. Even though the overall circuit topology is the same, the component choice will be cell-type dependent.

Controlling the response of the endogenous immune system is one of the frontiers of immune cell engineering. Yet, the endogenous immune systems represent a double-edged sword for CAR-T-cell therapy designers. A main adverse effect of CAR-T-cell therapy is cytokine release syndrome, which is provoked through endogenous macrophage activation by CAR-T cells[43–45].

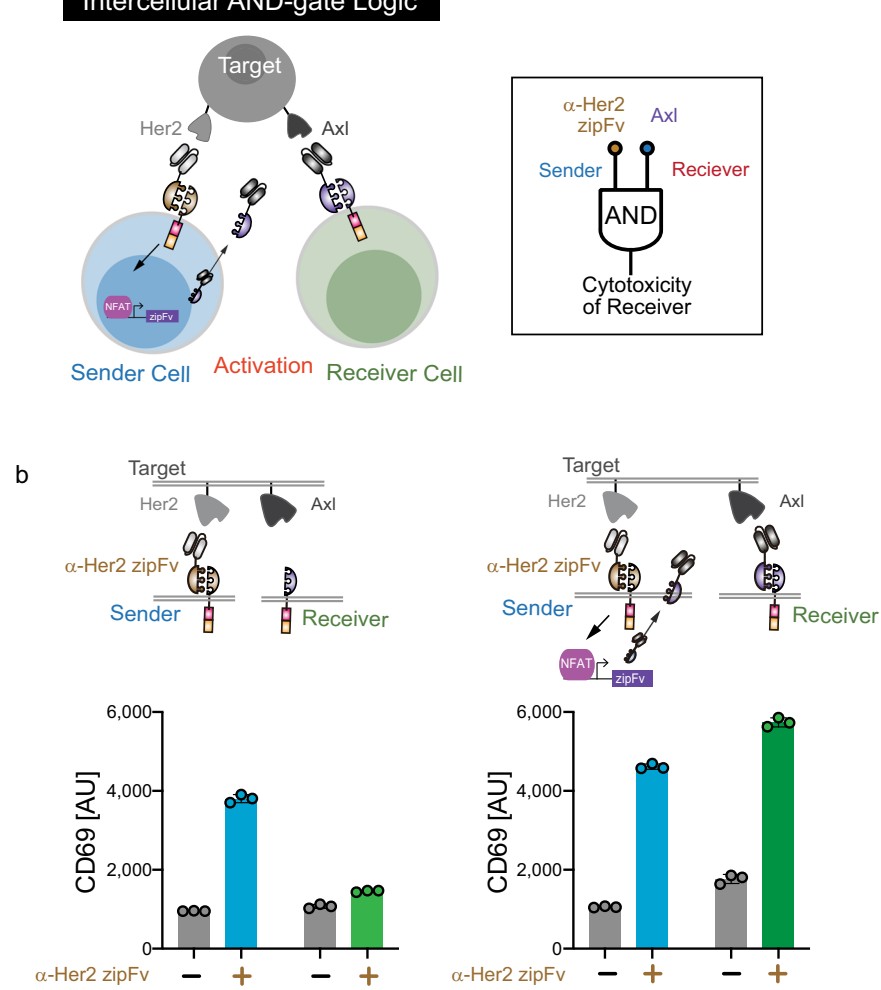

**Fig. 7 The intercellular AND gate using zipFv secretion system. a** Diagram of intercellular AND gate logic. CD4+ Sender cells express SYN6 CAR and secrete α-Axl-SYN2 zipFv when activated by α-Her2 zipFv. CD4+ receiver cells get activated by α-Axl-SYN2 zipFv secreted by sender cells. **b**. (Left) Addition of α-Her2 zipFv will activate sender cells that contain zipFv secretion module and secrete α-Axl-SYN2 zipFv, which will activate receiver cells as measured by CD69 expression level (Right) Activation of sender cells without zipFv secretion module does not lead to activation of receiver cells ($n = 3$, data are represented as the mean + SD).

On the other hand, CAR-T cells engineered to overexpress cytokines and chemokines that induce or attract myeloid cells have shown enhanced activity against solid tumors[46,47]. We demonstrated that SUPRA CAR-mediated activation of Th1 and Th2 cells induced macrophage polarization to the M1 (proinflammatory) and M2 (anti-inflammatory) state, respectively (Fig. 2). Existing data suggest that the temporal regulation of macrophage polarization can act as a negative feedback loop during the acute inflammation (first M1 inflammatory followed by M2 anti-inflammatory response)[48]. Therefore, the SUPRA CAR system has the potential to create a sequential order of inflammatory status through the temporal induction of helper T-cell activation (Th1 before Th2), which may help to reduce cytokine release syndrome without dampening the initial anti-tumor response.

In contrast to logic gates in the single cell, the distributed logic circuits comprised of multiple cell types would be applicable to therapies for multiple diseases. For instance, the A AND NOT (B AND C) circuit described in Fig. 5c could be repurposed without further engineering in different applications inside the patient. The CD4+ SUPRA CAR-T cells can be used for anticancer and the Treg SUPRA CAR can be used autoimmune disease at separate occasions simply by changing the zipFv addition. Therefore, our synthetic immune cell consortia is a multi-functional prosthetic immune system. Moreover, the sophistication of our synthetic cell consortia could be further enhanced with the addition of orthogonal inducible CAR system. We have shown that the α-FITC-CAR is compatible with our SUPRA CAR system Fig. 6e–g. The synthetic immune cell consortia represent another dimension in synthetic biology and cellular therapy.

## Methods

**zipCAR receptor construct design**. zipCARs were designed by fusing different leucine zippers[10] to the hinge region of the human CD8α chain and transmembrane and cytoplasmic regions of the signaling domains including co-stimulatory domains (CD28 and 4-1BB); co-inhibitory domains (PD-1, LAG3, TIM3, BTLA, and CTLA-4); and NK activating domains (2B4, DAP12, and NKG2D). They were under SFFV promoter for all primary T cells and NK cells experiments. All zipCARs contain surface tags and fused to fluorescent proteins or puromycin resistant gene to verify their expression.

**zipFv construct design**. The general design of zipFv is as follows. scFv (α-HER2, α-Axl, α-CD19, α-MESO, and α-CD5) is linked by a 35-aa glycine/serine linker to a

leucine zipper. Constructs were cloned into pSecTag2A vectors (Thermo Fisher Scientific) for transient expression. These vectors contain the CMV promoter, murine Ig-k-chain leader sequence, C-terminal c-myc epitope, and a 6X His tag for purification.

**Expression and purification of zipFv**. For transient expression of the protein, Freestyle 293-F cells (Thermo Fisher Scientific #R79007) were transfected with pSecTag2A plasmid according to the supplier's protocol. After 4 days of culture, cells were pelleted by centrifugation at $300 \times g$ for 5 min, and supernatant protein expression was confirmed by Coomassie gel stain (Thermo Fisher Scientific #24592) and western blotting with anti-Myc antibody (Abcam #ab62928). Proteins derived from transient transfection were purified as follows. Supernatant was passed through columns containing ProBond nickel chelating resin (Thermo Fisher Scientific #R80101). Then, each column was washed four times with native purification buffer (50 mM NaH2PO4 and 0.5 M NaCl pH 8.0) plus 20 mM imidazole (Sigma–Aldrich #I5513), and then eluted three times with native purification buffer plus 250 mM imidazole concentrations. Eluted proteins were concentrated to ~2 mL and dialyzed into 1× PBS (Thermo Fisher Scientific #AM9625). After dialysis, the protein was verified by western blot and SDS-PAGE gel electrophoresis and protein concentration was quantified by the Pierce BCA Protein Assay Kit (Thermo Fisher Scientific #23227).

**Primary human T cells isolation and culture**. Anonymized and deidentified normal whole peripheral blood was obtained from Boston Children's hospital as approved by Boston University Institutional Review Board (IRB). Primary human CD4+ and CD8+ T cells were isolated from anonymous healthy donor blood by negative selection (STEMCELL Technologies #15062 and #15063). Purities of CD4+ and CD8+ T cells were checked with FITC Mouse Anti-Human CD4 (1:200 dilution, BD, Clone RPA-T4) and Pacific Blue™ Mouse Anti-Human CD8 (1:200 dilution, BD, clone RPA-T8), respectively. T cells were cultured in human T-cell medium consisting of X-Vivo 15 (Lonza #04-418Q), 5% Human AB serum (Valley Biomedical #HP1022), 10 mM N-acetyl L-Cysteine (Sigma–Aldrich #A9165), 55 µM 2-mercaptoethanol (Thermo Scientific #31350010) supplemented with 50 units/mL IL-2 (NCI BRB Preclinical Repository). T cells were cryopreserved in 90% heat-inactivated FBS and 10% DMSO.

Regulatory T cells (Tregs) were isolated using immunomagnetic cell isolation kit (STEMCELL Technologies #18063 or #17861) and purity was checked with Alexa Fluor® 647 Mouse anti-Human FoxP3 Clone 259D/C7 (1:100 dilution, BD Bioscience, #560045), BV510 Mouse Anti-Human CD25 Clone 2A3 (1:100 dilution, BD Bioscience, #740198), or BV711 Mouse Anti-Human CD25 Clone 2A3 (1:100 dilution, BD Bioscience, #563159) They were cultured initially in human T-cell medium consisting of X-Vivo 15, 5% Human AB serum, 10 mM N-acetyl L-Cysteine, 55 µM 2-mercaptoethanol supplemented with 200 units/mL IL-2. N-acetyl L-Cysteine and 2-mercaptoethanol were removed during the Treg suppression experiment. Gamma delta (γδ) T cells were isolated using immunomagnetic negative selection cell isolation kit (STEMCELL Technologies #19255) from whole blood. Purity was checked with APC anti-human TCR Vδ2 Antibody (1:100 dilution, Biolegend, clone B6) and Brilliant Violet 421 anti-human TCR α/β Antibody (1:100 dilution, Biolegend, #306722). Purified γδ T cells were activated with Zoledronic acid 3 µg/mL (Sigma–Aldrich #1724827). After 5 days of activation, γδ T cells were transduced with lentivirus as shown below.

**Primary NK cell isolation and culture**. Primary NK cells and apheresis cells were obtained from Senti Biosciences through overnight shipment. NK cells were isolated using the Human NK Cell Isolation Kit (Miltenyi Biotech, 130-092-657) via autoMACS system (Miltenyi Biotech). The feeder cells were prepared as described before[20] by irradiation of the apheresis cells from the same donor with NK cells we used. In short, the apheresis cells were irradiated by MultiRad 350 (Precision X-ray) with 20 Gy under SnCuAl filter, and NK cells were mixed with the feeder cells at NK: feeder cells ratio of 1:5. The cell mixture was cultured in NK MACS media (Miltenyi Biotech, #130-114-429) supplemented with 5% human AB serum, 500 IU/mL of human IL-2, and 10 ng/mL of OKT-3 (Thermo Fisher, #14-0037-82). The population of NK cells and T cells were checked with anti-CD3-Alexa Fluor 647 antibody (1:100 dilution, Biolegend, #300322) and anti-CD56-APC-Cy7 antibody (1:100 dilution, Biolegend, #362511), and purified NK cells were maintained over 98% since day 7 of the co-culture.

**Lentiviral transduction of human T cells and NK cells**. Replication-incomplete lentivirus was packaged via transfection of HEK293FT cells (Thermo Fisher Scientific #R70007) with a pHR transgene expression vector and the viral packaging plasmids: pMD2.G encoding for VSV-G pseudotyping coat protein (Addgene #12259), pCMVR8.74 (Addgene#22036), and pAdv (Promega). One day after transfection, viral supernatant was harvested every day for 3 days and replenished with prewarmed Ultraculture media (Lonza #12-725 F) with 2 mM L-glutamine, 100 U/mL penicillin, 100 µg/mL streptomycin, 1 mM sodium pyruvate, and 5 mM sodium butyrate. Then, the harvested virus was purified through ultracentrifugation or concentrated with Lentivirus concentrator (Takara #631232). One day before transduction, T cells (CD4+, CD8+) were stimulated with Human T-activator CD3/CD28 Dynabeads (Thermo Fisher Scientific #11132D) at a 1:2 cell:

bead ratio and cultured for 24 h. After viral supernatant purification or concentration, retronectin (Clontech #T100B) was used to transduce cells. Briefly, non-TC-treated 6-well plates were coated with retronectin following the supplier's protocol. Then, concentrated viral supernatant was added to each well and spun for 90 min at $1200 \times g$. After centrifugation, viral supernatant was removed and 4 mL of previously activated human T cells, primary NK cells or NK-92MI cells were added. Cells were spun at $1200 \times g$ for 60 min and moved to an incubator at 37 °C. For the selection of T cells and NK cell transduced with constructs having puromycin resistant gene cassette, 2 µg/mL puromycin was added to the culture media from day 3–6 after transduction. Transduction efficiency was measured with mCherry and GFP (if conjugated to CARs) or staining corresponding surface protein tags (Myc, Alexa Fluor 647 anti-Myc Tag (1:100 dilution, Cell Signaling Technology) or Alexa Fluor 405 Anti-c-Myc (1:100 dilution, Novus bio); V5, Alexa Fluor 647 anti-V5 Tag (1:100 dilution, R&D systems).

**Primary human Th1 and Th2 cells differentiation**. Primary human naïve CD4+ were isolated from anonymous healthy donor peripheral blood (STEMCELL Technologies #19555). After naïve CD4+ T-cell isolation, Th1 and Th2 cells were differentiated using the supplier's protocol (R&D cat. #CDK001 and #CDK002). Briefly, naïve CD4+ T cells were activated with Human T-cell activator CD3/CD28 dynabeads at a 1:2 cell:bead ratio and cultured for 24 h with Th1/Th2 differentiation media. One day after T-cell activation, primary human T cells were transduced with the methods mentioned above. During culture, T cells were cultured in T-cell differentiation media (Th1 or Th2 depending on the experiment) for at least 14 days.

**Cancer cell lines**. K562 myelogenous leukemia cells (ATCC #CCL-243), Jurkat T cells, NALM6 B cell precursor leukemia (ATCC #CRL-3273), THP-1 (kindly gifted from Siggers lab, Boston University) were cultured in RPMI-1640 (Lonza#12-702Q) with 5% (v/v) heat-inactivated FBS, 2 mM L-glutamine, 100 U/ mL penicillin and 100 µg/mL streptomycin. NK-92-MI (ATCC #CRL-2408) was cultured in X-Vivo 10 (Lonza #04-380Q) with 5% Human AB serum, 2 mM L-glutamine, 100 U/mL penicillin and 100 µg/mL streptomycin. Jurkat and NALM6 cells were electroporated with the PiggyBac Transposon system (System biosciences) to stably expression of surface antigens. THP-1 cells were transduced to express SUPRA CAR. Two days after transfection or transduction, antibiotic (Puromycin (Thermo Fisher Scientific #A1113803), zeocin (Thermo Fisher Scientific #R25005), or Hygromycin B (Thermo Fisher Scientific #10687010)) was added to the medium or FACS sorted to select for cells that express the transgenes. THP-1 cells were transduced using lentivirus to stably express zipCAR.

**Cytokine release assays**. Primary T cells expressing zipCAR were incubated with target cells ($10 \times 10^4$ cells/well) at an E:T ratio of 2:1 or 1:1 with corresponding zipFvs. After 24 h, the supernatant was harvested and followed the supplier's protocol to determine cytokine release level (IFN-γ (BD Biosciences #555142), IL-4 (BD Biosciences #555194)).

**Staining activation markers**. Engineered T cells were stained with APC-conjugated α-CD69 antibody (1:200 dilution, BioLegend #310910) 24 h after starting co-culture. Engineered Treg cells were stained with FITC-conjugated α-CTLA-4 antibody (1:50 dilution, Thermo Fisher Scientific #11-1529-42) 48 h after starting co-culture. Expression levels of activation markers in zipCAR+ cells were measured by flow cytometry (Attune-NxT, Thermo Fisher Scientific) and analyzed using FlowJo 10 (TreeStar).

**Cytotoxic assay**. Cytotoxicity assays were carried out using the flow cytometer (Except for Fig. 1a). Briefly, CAR-T cells were co-cultured with zipFv and target cells for 24 h at 37 °C. Prior to flow cytometry, control samples containing only the target cells were used to set a flow cytometry gate for intact target cells based on forward and side scatter patterns that had been previously confirmed to exclude dead cells (Supplementary Fig. 3e). Also, fluorescent proteins and staining CD19 (anti-Human CD19 PE-Cy7 (1:200 dilution, Tonbo Bioscience, clone HIB19)) were used to further identify the target cells, or staining CD8 (BV421 Mouse Anti-Human CD8 (1:200 dilution, BD)) was used to exclude CAR-T cells This gate was applied to all samples. Live target cell number was calculated and target cell cytotoxicity was calculated using the following formula: Cytotoxicity = 100 × [(Total live target cell number − number of remaining live cells after lysis)/(Total live target cell number)].

**Luciferase cytotoxic T lymphocyte assay**. Cytotoxicity assays was carried out using bioluminescence to generate Fig. 1a. Briefly, CAR-T cells were incubated with zipFv and target cells that were engineered to express luciferase at varying effector to target ratio for 4 h at 37 °C. Initially, target cells were seeded at 75,000 or 100,000 cells per well (96-well plate) and zipFv at varying concentrations were added (amount of zipFvs were titrated to give maximum response). Then, engineered T cells were added (unless otherwise noted, T cells used in the experiment were not sorted based on the SUPRA CAR expression level). After ~4 h incubation, culture medium was removed to leave 50 ul per well, then 50 ul of prepared luciferase

reagent (Promega #E2610) was added to each well of the 96-well plate (Corning #3904). Measurements were performed with the SpectraMax M5 (Molecular Devices). Target cell cytotoxicity was calculated using the following formula: Cytotoxicity $= 100 \times$ [(Total Target cell luminescence − luminescence of remaining cells after lysis)/(Total Target cell luminescence)].

**Treg cell suppression assay.** Effector T cells were stained with CellTrace Violet dye (Thermo Fisher Scientific #C34557) following the manufacturer's instructions. NALM6 target cells expressing Her2 and Axl were treated with 3 μg/mL mitomycin C (Sigma–Aldrich #M4287) to make target cells replication-incompetent. Treg cells and Teff cells were mixed at 1:1 or 2:1 ratio with target cells. zipFvs were also added to final concentrations. Cells were collected for flow cytometry analysis after incubation for 4 days.

**Phagocytosis assay.** Two days before phagocytosis assays, zipCAR-containing THP-1 cells (kindly gifted by Dr. Siggers at Boston University) were activated with Phorbol 12-Myristate 13-Acetate (PMA; Fisher Scientific) with 25 ng/mL and seeded into TC-treated 96-well plates ($15 \times 10^4$ cells/well). One day after activation, media in each 96 wells was replaced with new media without PMA and incubated at 37 °C. After 24 h, NALM6 target cells were stained with CellTrace violet dye following manufacturer's instructions and added into each well ($15 \times 10^4$ cells/well) with the corresponding zipFv. Cells were spun down by centrifugation at $400 \times g$ for 1 min. 4 h after co-culture, cells were treated with 5 mM EDTA (Fisher Scientific) in PBS for 15 min at 4 °C and stained with APC anti-human CD44 antibody (1:100 dilution, Biolegend #338806) to label THP-1 macrophages and cells were collected for flow cytometry analysis. Prior to flow cytometry, control samples containing only the target cells were used to set a flow cytometry gate for intact target cells based on forward and side scatter patterns that had been previously confirmed to exclude apoptotic cells. Also, fluorescence markers were used to further identify the target cells. The number of phagocytosed cells were calculated by dual positive cells (APC + [α-human CD44] and violet + [CellTrace]). This gate was applied to all other samples. Target cell phagocytosis was calculated using the following formula: % phagocytosis $= 100 \times$ [(Number of THP-1 macrophage that are dual positive after co-culture)/(Total live THP-1 macrophage cell number)].

**M1 and M2 macrophage polarization assay.** In TC-treated 96-well plates, THP-1 cells were treated with 25 ng/mL PMA for 6 h to differentiate macrophages and then cultured for 2 days in new media without PMA. Antigens-expressing Nalm6 ($1 \times 10^5$ cells/well), RR-CAR-expressing Th1 cells ($1 \times 10^4$ cells/well), and FOS-CAR-expressing Th2 cells ($2.5 \times 10^4$ cells/well) were added to THP-1 with or without corresponding 4 nM zipFvs. THP-1 cells were detached with 5 mM EDTA in PBS 24 h after starting co-culture. THP-1 cells were stained with antibodies against M1 and M2 markers (APC anti-HLA-DR (1:100 dilution, BioLegend #307609), APC anti-CCR7 (1:100 dilution, BioLegend #353214), and PE anti-CD206 (1:100 dilution, BioLegend #321105)) and analyzed by flow cytometry.

**Kill switch assay.** Nalm6 target cells ($10 \times 10^4$ cells/well), SYN6-V5 zipCAR CD8 + T cells ($5 \times 10^4$ cells/well; transduction efficiency was approximately 50%), and α-FITC-CAR-NK-92MI cells ($5 \times 10^4$ cells/well) were co-cultured with 4 nM α-Her2-SYN5 zipFv and/or 500 ng/mL FITC-conjugated α-V5 tag antibody (Thermo Fisher Scientific #R963-25). Twenty-four hour later, live target cells and CD8+ T cells were measured by flow cytometry.

**zipFv secretion intercellular AND gate logic assay.** CD4+ T cells were lentivirally transduced with both zipCAR and NFAT-zipFv secretion module (contains puromycin resistant gene). After transduction, puromycin was applied to the transduced cell at 2 μg/mL concentration for 10 days. Both SUPRA CAR-expressing CD4+ and CD8+ cells were as receiver cells. Nalm6 target cells ($10 \times 10^4$ cells/well), CD4+ sender cells ($10 \times 10^4$ cells/well), and CD4+/CD8+ receiver cells ($5 \times 10^4$ cells/well) were co-cultured with α-Her2-SYN5 zipFv for 2 days. CD69 expression was measured afterward.

**Human cancer xenograft mouse model.** Female NSG mice, 4–6 weeks of age, were purchased from Jackson Laboratories (#005557) and maintained in the BUMC Animal Science Center (ASC). All protocols were approved by the Institutional Animal Care and Use Committee at BUMC. In order to carry out the intraperitoneal xenograft models, NSG mice were initially injected with $0.5 \times 10^6$ luciferase-expressing SKOV3 (Cell Biolabs AKR-232) intraperitoneally at day 0. At day 1 and day 8, $10 \times 10^6$ CAR-NK cells were infused intraperitoneally twice (total $20 \times 10^6$ cells). Four hundred μg/kg of α-Her2-EE zipFv and 4 mg/kg of α-Her2-SYN9 zipFv or α-Her2 scFv (Mock) were added intraperitoneally every day from day 1 until day 12, and then 0.4 mg/kg of α-Her2-EE zipFv and 1 mg/kg of α-Her2-SYN9 zipFv or α-Her2 scFv (Mock) were added intraperitoneally every day from day13 until day 21. Tumor burden was measured by IVIS Spectrum (Xenogen) and was quantified as total flux (photons per sec) in the region of interest. Images were acquired within 10 min following intraperitoneal injection of 150 mg/kg of D-luciferin (PerkinElmer #122799). Images were analyzed using Living Images (Perkin Elmer).

**Statistical analysis and reproducibility.** Statistical significance was determined by student's $t$-test (two-tailed) unless otherwise noted. All curve fitting was performed with Prism 7 (Graphpad) and heat maps were drown with Microsoft Excel. $p$ values are reported (not significant $= p > 0.05$, $*p \leq 0.05$, $**p \leq 0.01$, $***p \leq 0.001$). All error bars are represented either SEM or SD.

Each experiment was repeated at least two times with more than two technical replicates. Experiment with single biological replicate was in vivo experiment (Fig. 4f–g) and γδ T-cell experiment (Fig. 1e).

**Reporting summary.** Further information on research design is available in the Nature Research Reporting Summary linked to this article.

## Data availability
Reagent requests should be directed and will be fulfilled by the lead author Wilson Wong (wilwong@bu.edu). Any other relevant data are available upon reasonable request. Source data are provided with this paper.

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

## Acknowledgements

J.H.C. acknowledges funding from the Kwanjeong Educational Foundation. A.O. is supported by The Uehara Memorial Foundation Overseas Research Fellowship, The Nakatomi Foundation, and The Kanae Foundation for the Promotion of Medical Science. J.J.C. acknowledges funding from the Paul G. Allen Frontiers Group and the Wyss Institute. W.W. acknowledges funding from the Boston University Ignition Award, NIH (1DP2CA186574, 1R56EB027729-01A1, 1R01GM129011-01, R01EB029483), NSF Expedition in Computing (1522074), NSF CAREER (162457), NSF BBSRC (1614642) and sponsor research agreement from Senti Biosciences. We thank Teresa Wiese for cloning zipCAR constructs used for Jurkat T-cell experiments and Devin R. Burrill for initial guidance on protein purifications. We also thank Wong lab members for suggestions on the manuscript; Dr. Jennifer Cappione and Dr. Anna Belkina from the BU flow cytometry core for flow cytometry assistance; Dr. Thomas Balon and Dr. Francesca Seta from the BU animal facility for IVIS imaging assistance.

## Author contributions

J.H.C. and A.O. designed and generated genetic constructs, performed experiments, and generated figures. K.S. helped to perform zipFv secretion experiment and to generate related genetic constructs. S.L. performed the primary NK experiments and NOT logic experiment with BTLA. J.H.C., A.O., J.J.C., and W.W. analyzed data. J.H.C., A.O., and W.W. created figures. W.W. conceived the project. J.H.C., A.O., J.J.C., and W.W. wrote the paper. All authors commented on and approved the paper.

## Competing interests

The authors declare the following competing interests: Boston University has filed a patent application, WO2017091546A1, (Methods and compositions relating to chimeric antigen receptors) with J.H.C. and W.W.W. as the named inventor based on this work. W.W.W. is a co-founder and shareholder of Senti Biosciences, and received research support from Senti Biosciences. J.J.C. is a co-founder and shareholder of Senti Biosciences. A.O. is a current employee of Hitachi, Ltd. J.H.C. is a current employee of Spark Therapeutics. K.S. is current employee of KSQ Therapeutics. All other authors declare no competing interests.
