## [Peer Review File · Nature Communications]

Reviewers' Comments:

Reviewer #2:

Remarks to the Author:

Manuscript background information

Cho et al. describe an extension of their groups previous work (Cho JH, Cell. 2018) developing split, universal, and programmable (SUPRA) chimeric antigen receptors (CAR). The SUPRA CAR system is a split CAR system in which a soluble extracellular antigen binding scFV portion (zipFV) is split from the transmembrane/intracellular portion (zipCAR). Pairing of the two components is dependent on the dimerization of compatible leucine zipper domains. This strategy offers potential advantages over traditional CAR designs including the ability to modulate targeting, fine-tune signaling/activation, and incorporate basic logic and control functions. In the current manuscript, Cho et al. extend SUPRA CAR application by first validating its function in more diverse immune cell subsets including additional subsets of T cells (Treg, Th1, Th2, and gamma delta), as well as NK- and macrophage-like cells. The authors show data that orthogonal zipCARs can be used to differentially activate Th1 and Th2 T cells, which in turn can regulate polarization of macrophages in vitro. To address current issues of CAR antigen specificity the authors demonstrate tunable AND gate function with zipCARs in CD4 T cells and T regs, which is accomplished by separating CD3zeta and co-stimulatory signaling on separate zipCARs. A NOT gate function is achieved by incorporating an orthogonal zipCAR with a negative regulatory domain. This function is demonstrated in an NK92 cell line both in vitro and in vivo. Using simultaneous delivery of three orthogonal zipCARs the authors demonstrate that the AND/NOT functions can be combined to allow A AND B (NOT C) within a single cell, as well as A AND (NOT B AND C) logic in a system that involves specific activation of Treg and consequent suppression of activated CD4s. Also described is a kill switch mechanism involving ADCC of CAR T cells by engineered NK92. Finally, by incorporating conditional activation and secretion of zipFVs in "sender cells", the authors demonstrate that activation of a second CAR function in "receiver cells" can be made dependent upon initial activation by sender cells. The overall thrust of the manuscript is that incorporation of the SUPRA CAR system in diverse immune cells can be used to coordinate and control immune function in a way that has not been possible previously.

Comments for the authors

The current study extends upon this groups previous development of the SUPRA CAR system (Cho, JH. et al. Cell 2018) and demonstrates functional execution of applications theorized in their initial work. Overall the manuscript is well written, and the applications demonstrated are highly relevant to immune therapy. However, there are several key limitations that if addressed would significantly improve the quality of the study.

Major limitations:

1. A major limitation is that many of the functional assays are superficial, often consisting of a single readout, i.e. IFN γ secretion or an activation marker. Each figure has a very nice diagram of what is predicted to be happening, but the supporting data is not comprehensive nor fully convincing. Much more data could easily be obtained in these experiments to support conclusions. For instance, what is happening on a cell by cell basis? This could be tested using intracellular flow cytometry, where multiple cytokines, activation markers, and zipCAR expression can be explored simultaneously to fully grasp the robustness of activation. For example, when tuning the activation response by dosing zipFv, separating activation domains, or including inhibitory receptors, is the change in cytokine secretion due to 50% of the cells no longer responding while half still respond at 100%? Or do all of the cells decrease by 50%. This is important from a therapeutic perspective.
2. The only truly primary immune cells used in this study are T cells. Both the NK and macrophage studies were done using the NK-92 and THP-1 cell line representatives. Some demonstration of function in primary NK and monocyte/macrophage would raise the impact substantially.

3. In the systems where 3 zipCARs are delivered to cells, the proportion of each cell that expresses all three is unclear. Puromycin is used to enrich on a single construct in some cases, and Fig S3 has flow plots for two fluorescent markers, but there is no data to indicate the representative distribution of the multiple zipCARs in the population. This seems like it would be important as proper function relies on all signals being present.

4. In the NOT logic experiments using NK-92, it is confusing as to why the in vitro positive and negative signals are linked to different targets. Why is Her-2 used as both the positive and negative antigen in vivo, when in vitro Her-2 is positive and Axl-2 is the negative?

Minor comments

1. Citation should be included for claim that gamma delta T cells do not cause GvHD.

2. Published protocols for zoledronate expansion of gdT cells rely on stimulation with other mononuclear cells as they produce the appropriate phosphoantigens for stimulation, but in this manuscript the gdT cells are purified and then stimulated with zol. Is this a mistake or is there a publication for this strategy?

3. In the methods for cytotoxicity assay it is stated that a bioluminescence assay is used, but then the methods describe a flow based assay? Further, the flow assay uses only forward and side scatter to discriminate. Inclusion of a cleaved caspase readout or at least a viability dye would be more convincing.

Reviewer #3:

Remarks to the Author:

Cho et al., explore the versatility of their SUPRA CAR system to engineer advanced logic and cell-to-cell communication in various primary immune cell types, including T cells, natural killer cells and macrophages. The system is designed such that a leucine zipper in the extracellular domain of the SUPRA CAR specifically binds to a soluble zipper fused to an antibody fragment specific to an antigen on a target cell. In the current work, the authors expand the potential of SUPRA CAR by creating an inhibitory feature that allows to generate a NOT-gate and 3-input (A AND NOT (B AND C)) logic circuits in a single cell, as well as inducible intercellular AND-gates, NOT-gates and kill switches between different immune cell types, thereby generating immune cell consortia with potential to treat cancer and autoimmune diseases. Overall, this is a well conducted study with sound data on the in vitro performance of the differently engineered immune cells, as well as of the in vivo performance of NOT-gate-engineered natural killer cells in a mouse cancer model. Therefore, I recommend its publication in Nature Communications once the following minor points are addressed:

- From Figure 1, it seems that all immune cell types were engineered with the RR ZipCAR except the gamma-delta T cells, which express the FOS ZipCAR. Was there any reason for that? Please clarify.
- It could be useful to provide the percentage of cells expressing the ZipCARs. The flow cytometry plot in supplementary Fig. 1e shows mcherry positive cells but it is not mentioned from which cells. Was the lentiviral transduction efficiency similar for all immune cell types tested? The multiplicity of infection (MOI) used is also missing.
- Please revise sentence "Next day, we infused CD3/BTLA NOT gate expressing NK cell at day 1 and day 8."

Summary of the revision

We first want to thank the reviewers for their overall positive and enthusiastic response to our work and the thoughtful and constructive comments on our results. One of the primary criticisms of our work was that while we have demonstrated SUPRA CAR in many different primary immune cells, they are derived from the T cell lineage. Therefore, we now have collected and presented experimental data showing that the SUPRA CAR is also functional in human primary NK cells (**Fig S1f** of the revised paper).

Reviewer #2 (Remarks to the Author):

Comments for the authors

The current study extends upon this groups previous development of the SUPRA CAR system (Cho, JH. et al. Cell 2018) and demonstrates functional execution of applications theorized in their initial work. Overall the manuscript is well written, and the applications demonstrated are highly relevant to immune therapy. However, there are several key limitations that if addressed would significantly improve the quality of the study.

Thank you for the positive comments. We have substantially revised the manuscript to address the constructive comments provided.

Major limitations:

1. A major limitation is that many of the functional assays are superficial, often consisting of a single readout, i.e. IFN γ secretion or an activation marker. Each figure has a very nice diagram of what is predicted to be happening, but the supporting data is not comprehensive nor fully convincing. Much more data could easily be obtained in these experiments to support conclusions. For instance, what is happening on a cell by cell basis? This could be tested using intracellular flow cytometry, where multiple cytokines, activation markers, and zipCAR expression can be explored simultaneously to fully grasp the robustness of activation. For example, when tuning the activation response by dosing zipFv, separating activation domains, or including inhibitory receptors, is the change in cytokine secretion due to 50% of the cells no longer responding while half still respond at 100%? Or do all of the cells decrease by 50%. This is important from a therapeutic perspective.

Thank you for your constructive comments. We agree that single-cell data, especially the cytokine production, will be important to understand how our system behaves. However, the main goal of our work is on the logical response of our system. Therefore, we focus on measuring cytotoxicity whenever possible because it is one of the most clinically relevant T cell responses. Nonetheless, we want to acknowledge, both here and in the manuscript, that single-cell cytokine data would also help determine the overall function and potential of the engineered immune cells. However, because of the large number of cell types and systems that we created, it is too expensive and time consuming to measure cell killing and all the cytokines at the single-cell level.

While we do not have the resources to recollect all the data with single-cell measurement, we do have CD69 histogram expression data, a commonly used T cell activation marker, for an AND gate in T cells (Fig 3b). We acknowledge that the CD69 data do not indicate how many different cytokines do the T cells produce. The data only indicate how many T cells are activated. As shown by the data, not all the cells were activated. We believe that not all the cells activated because not all the cells expressed both CARs at a sufficient level. While our AND gate clearly works logically as intended, the single-cell data provide critical information on how to optimize our system further. The CD69 data are included in Supplementary **Fig S3b** of the revised paper. We also provide a brief explanation and implication of the data in the discussion section, saying that our data only indicate the logical performance, and the full immune response remains to be explored.

2. The only truly primary immune cells used in this study are T cells. Both the NK and macrophage studies were done using the NK-92 and THP-1 cell line representatives. Some demonstration of function in primary NK and monocyte/macrophage would raise the impact substantially.

Thank you for your constructive comments. We have taken your comments seriously and have performed additional experiments with primary NK cells, which has proven to be much more challenging than working with primary T cells. After overcoming many issues with reagent supplies and COVID-related restriction, we were able to expand and transduce human primary NK cells with transduction efficiency of around 50%, although the average expression level is somewhat lower than other immune cells. Nonetheless, we have shown that our SUPRA CAR also works in primary NK cells. However, our primary NK cells have substantial basal killing against our NALM6 target cells, as demonstrated by almost 60% killing from untransduced primary NK cells. ZipCAR expression alone (without zipFv) also leads to 60% basal killing, indicating that zipCAR does not kill more cancer cells without zipFv. The addition zipFv increases cell killing to 80%. While the increase may seem modest, our results show that the SUPRA CAR can have inducible cell killing in primary NK cells. Further reduction of the basal killing, probably using a different cell line, and improved transduction to increase efficiency and expression level, will undoubtedly increase the target cell killing. These results are now included in **Supplementary Figure S1f** of the revised paper.

3. In the systems where 3 zipCARs are delivered to cells, the proportion of each cell that expresses all three is unclear. Puromycin is used to enrich on a single construct in some cases, and Fig S3 has flow plots for two fluorescent markers, but there is no data to indicate the representative distribution of the multiple zipCARs in the population. This seems like it would be important as proper function relies on all signals being present.

Thank you for your comment. We are sorry that we were not clearer before. Figure S3 is a flow cytometry data measuring the expression level of two receptors (one receptor is CD3 ζ and the other receptor is CD28 or 41BB) to demonstrate a two-input AND gate in T cells. For CD8+ T cells, the transduction of two receptors was efficient, and more than 50% of cells contain both receptors (GFP+mCherry+ cells, supplementary fig. 3a). For regulatory T cells, we used puromycin, myc, and V5 tag to gate for the cells containing two receptors. This is because

SYN6-V5-CD3z receptor also contains FOXP3 transcription factor using P2A. Due to the large DNA footprint of the FOXP3 transcription factor, transduction efficiency was lower (~36% of cells express both receptors). Yet, regulatory T cells also showed a synergistic effect from both receptors.

In the systems where 3 zipCARs are delivered, we first use puromycin to select for one receptor (e.g., SYN1-BTLA receptor). Then, we gate for the mCherry positive cells that contain two receptors (SYN6-CD28 and FOS-CD3 ζ , supplementary fig. 6b). From our previous experiment, we know that most of the mCherry positive cells are positive for both receptors. We have now added these data in **supplementary Fig. S6c and modified the main text**. As demonstrated from this figure, most of the mCherry positive cells are positive for both myc and V5 tag. Thus, in our three-input experiments, we believe at least 30% of cells are positive for all three SUPRA receptors.

4. In the NOT logic experiments using NK-92, it is confusing as to why the *in vitro* positive and negative signals are linked to different targets. Why is Her-2 used as both the positive and negative antigen *in vivo*, when *in vitro* Her-2 is positive and Axl-2 is the negative?

We are sorry for the confusion. We used α -Her2 zipFv for both the positive and negative antigen *in vivo* because it was costly to use α -Axl zipFv. The yield of α -Axl zipFv after protein purification was not high enough and therefore is very expensive to produce for *in vivo* mouse study. As the main purpose of the experiment was to demonstrate that we can modulate different signaling domains *in vivo* using the SUPRA platform, we decided to perform the experiment using zipFv targeting the same Her2 antigen.

Minor comments

1. Citation should be included for claim that gamma delta T cells do not cause GvHD.

Thank you for the comment. We now have included literature related to gamma delta T cells and GvHD and modified our main text.

2. Published protocols for zoledronate expansion of gdT cells rely on stimulation with other mononuclear cells as they produce the appropriate phosphoantigens for stimulation, but in this manuscript the gdT cells are purified and then stimulated with zol. Is this a mistake or is there a publication for this strategy?

We agree with the reviewer's comment that the mechanism of gamma delta T cell activation relies on the accumulation of isopentenyl pyrophosphate (IPP) on mononuclear cells, which acts as antigen-presenting cells. The commercial kit that we used for isolating the gamma-delta T cell was not perfect, meaning that it included other mononuclear cells. In order to further enrich for gamma delta T cells, we treated isolated cells with zoledronate. We observed several-fold expansion of gamma-delta T cells, and this was the main reason why we treated isolated gamma delta T cells with zoledronate.

3. In the methods for cytotoxicity assay it is stated that a bioluminescence assay is used, but

then the methods describe a flow based assay? Further, the flow assay uses only forward and side scatter to discriminate. Inclusion of a cleaved caspase readout or at least a viability dye would be more convincing.

Thank you for your comment. We initially used bioluminescence assay but later changed to flow-based assay because we can measure cytotoxicity while also measuring T cell activation (CD69) and other extracellular markers at single-cell level. We performed 7AAD staining to validate our gating and validated that dead-cell staining did not affect live cell count. We now have included these data in **supplementary Fig. 2c** of the revised paper and modified our cytotoxicity assay method.

Reviewer #3 (Remarks to the Author):

Cho et al., explore the versatility of their SUPRA CAR system to engineer advanced logic and cell-to-cell communication in various primary immune cell types, including T cells, natural killer cells and macrophages. The system is designed such that a leucine zipper in the extracellular domain of the SUPRA CAR specifically binds to a soluble zipper fused to an antibody fragment specific to an antigen on a target cell. In the current work, the authors expand the potential of SUPRA CAR by creating an inhibitory feature that allows to generate a NOT-gate and 3-input (A AND NOT (B AND C)) logic circuits in a single cell, as well as inducible intercellular AND-gates, NOT-gates and kill switches between different immune cell types, thereby generating immune cell consortia with potential to treat cancer and autoimmune diseases. Overall, this is a well conducted study with sound data on the in vitro performance of the differently engineered immune cells, as well as of the in vivo performance of NOT-gate-engineered natural killer cells in a mouse cancer model. Therefore, I recommend its publication in Nature Communications once the following minor points are addressed:

We very much appreciate the reviewer's supportive comments.

- From Figure 1, it seems that all immune cell types were engineered with the RR ZipCAR except the gamma-delta T cells, which express the FOS ZipCAR. Was there any reason for that? Please clarify.

There was no specific reason for engineering gamma-delta T cells with FOS zipCAR. We initially wanted to demonstrate intercellular communication with gamma-delta T cells with other immune cell types that we engineered. Because FOS zipCAR and RR zipCAR are orthogonal to each other, we chose FOS zipCAR for gamma-delta T cells. Based on our experience, we are confident that RR zipCAR will also function in gamma-delta T cells.

- It could be useful to provide the percentage of cells expressing the ZipCARs. The flow cytometry plot in supplementary Fig. 1e shows mcherry positive cells but it is not mentioned from which cells. Was the lentiviral transduction efficiency similar for all immune cell types tested? The multiplicity of infection (MOI) used is also missing.

We apologize for the confusion in our original manuscript. We initially showed CAR expression in gamma delta T cells. Lentiviral transduction efficiency was similar to other immune cell types. Now we have included these data in **supplementary figure 1E** of the revised paper to show lentiviral transduction efficiency for different immune cell types that we tested.

- Please revise sentence "Next day, we infused CD3/BTLA NOT gate expressing NK cell at day 1 and day 8."

We have changed the main text to the following:

"After tumor engraftment, we infused CD3 ζ /BTLA NOT gate expressing NK cells at day 1 and day 8."

Reviewers' Comments:

Reviewer #2:

Remarks to the Author:

I would like to thank the authors for their efforts to address my critiques. These are challenging times and the effort made to conduct additional experiments in primary human NK cells is commendable and significantly improves the overall impact of the manuscript. I have no further comments and recommend the manuscript for publication in Nature Communications.